# Establishing age and gender-specific serum creatinine reference ranges for Thai pediatric population

**Sakon Suwanrungroj**[1☯], **Parichart Pattarapanitchai**[2☯], **Sirinart Chomean**[3,4☯], **Chollanot Kaset**[3,4☯]*

**1** Queen Sirikit National Institute of Child Health, Thung Phayathai Subdistrict, Ratchathewi, Bangkok, Thailand, **2** Department of Statistics, Faculty of Science, Chiang Mai University, Chiang Mai, Thailand, **3** Department of Medical Technology, Faculty of Allied Health Sciences, Thammasat University, Pathum Thani, Thailand, **4** Thammasat University Research Unit in Medical Technology and Precision Medicine Innovation, Pathum Thani, Thailand

☯ These authors contributed equally to this work.
* Chollanotk@gmail.com

**Data Availability Statement:** All relevant data are within the manuscript and its Supporting Information files.

## Abstract

Accurate assessment of kidney function in children requires age and gender-specific reference ranges for serum creatinine. Traditional reference values, often derived from adult populations and different ethnic backgrounds, may not be suitable for children. This study aims to establish specific reference ranges for serum creatinine in the Thai pediatric population, addressing the gap in localized and age-appropriate diagnostic criteria. This retrospective study analyzed serum creatinine levels from Thai children aged newborn to 18 years, collected from the Laboratory Information System of the Queen Sirikit National Institute of Child Health from January 2017 to December 2021. The Bhattacharya method was employed to establish reference ranges, considering different age groups and genders. The study compared these newly established reference values with international studies, including those of Schlebusch H., Pottel H., and Chuang GT., to validate their relevance and accuracy. A total of 27,642 data entries (15,396 males and 12,246 females) were analyzed. The study established distinct reference ranges for serum creatinine, which varied significantly across different age groups and between genders. These ranges were found to gradually increase with age from 2 months to 18 years. The study also highlighted notable differences in reference values when compared with other ethnic populations. The study successfully establishes tailored reference ranges for serum creatinine in Thai children, providing a valuable tool for more accurate diagnosis and monitoring of kidney health in this demographic. This initiative marks a significant advancement in pediatric nephrology in Thailand and suggests a need for continuous refinement of these ranges and further research in this area.

## Introduction

Creatinine testing is a popular method for assessing kidney function, or glomerular filtration rate (GFR), due to its simplicity, speed, and affordability. Many laboratories report GFR values

**Funding:** Thammasat University Research Unit in Medical Technology and Precision Medicine Innovation. The funders had no role in study design, data collection and analysis, decision to publish, or preparation of the manuscript.

**Competing interests:** NO authors have competing interests.

alongside serum creatinine tests, as creatinine levels are a crucial factor in evaluating kidney function. Besides kidney function, other factors such as muscle mass and high meat consumption can also alter serum creatinine levels. These variations in creatinine levels can be influenced by age, gender, and ethnicity [1, 2] meaning similar serum creatinine levels could indicate different GFR levels. Therefore, serum creatinine interpretation must be done with caution, especially in children and the elderly, due to varying factors like age and muscle mass changes.

If a patient shows consistently rising creatinine levels reaching the upper limit, it could signal early-stage kidney abnormalities. In most patients with serum creatinine exceeding the upper reference limit, it is often found that kidney filtering efficiency has already decreased by 50% [3]. The upper reference limit for children is significantly lower than adults, and the reference range increases gradually with age [4]. Therefore, pediatric serum creatinine reference values should be specific to that population group.

Populations from infancy to adolescence undergo significant growth and muscle development, leading to distinct physical and physiological characteristics. Previous studies have highlighted the differences in creatinine reference ranges between children and adults [5]. Age and gender significantly impact blood creatinine levels, so reference values should be grouped accordingly for accurate disease diagnosis and treatment monitoring. Obtaining reference data for children can be challenging due to difficulties in blood sampling and ethical considerations. Thus, using hospital database information to establish reference values is an alternative. The Bhattacharya method, a statistical approach, is recommended for determining reference values from databases (the indirect method) [6]. This method helps separate overlapping data sets of healthy individuals and those with diseases [7].

Our primary objective is to establish a reliable reference range for serum creatinine levels specifically tailored to Thai children. This need arises from the prevalent practice in many laboratories of adopting reference values from adult populations and different ethnicities, as provided in company leaflets. Such generalized values are often unsuitable for pediatric cases, given the significant physiological differences between children and adults, as well as the variations across ethnic groups. By developing a dedicated reference range for Thai children, we aim to enhance the accuracy of kidney function assessments in this demographic, thereby facilitating more precise diagnoses and effective treatment strategies for pediatric patients.

## Materials and methods

### Samples

Serum creatinine levels from patients aged newborn to 18 years were collected using the Laboratory Information System [8] of the Queen Sirikit National Institute of Child Health, from January 1, 2017, to December 31, 2021. The data included only the first test for each individual. Exclusion criteria were patients with conditions affecting creatinine levels, including data from nephrology and diabetes clinics, and samples with inappropriate physical characteristics [9–11]. The age groups were divided into ten categories as follows: (newborn—<2m), (2m-12m), (1- <3y), (3- <5y), (5- <7y), (7- <9y), (9- <11y), (11- <13y), (13- <15y), (15-18y), based on Schlebusch H.'s study [12]. Each age group was further divided into male and female, making a total of 20 groups. The sample size for establishing reference values was determined using the Indirect method as recommended by The International Federation of Clinical Chemistry (IFCC) Committee on Reference Intervals and Decision Limits (C-RIDL). Each divided group contained no fewer than 400 data points. The process of establishing reference values from the Queen Sirikit National Institute of Child Health's database using the Bhattacharya method is depicted in Fig 1.

## Instruments, reagents, and quality control

Serum creatinine was analyzed using the COBAS INTEGRA 400/400 PLUS (Roche Diagnostic GmbH, Germany) automated analyzer, utilizing the enzymatic colorimetric method. The reagent used was Creatinine plus ver.2 (CREP2) from Roche Diagnostics (Thailand) Ltd. The analyzer's calibrator was calibrator f.a.s. Internal laboratory quality control was maintained using PreciControl ClinChem multi 1 and 2, and external quality control was performed through the External Quality Assurance Services, Clinical Chemistry (Monthly) Program, BIO-RAD. Furthermore, the operational process from pre-analytical, analytical, and post-analytical stages has been enhanced and certified for laboratory quality based on the ISO/FDIS 15189:2000 standards, as well as for hospital and healthcare service quality, certified by the International Society for Quality in Health Care External Evaluation Association (ISQua EEA). This ensures adherence to standards and verifiability at every stage, including personnel expertise, equipment readiness, calibration, and maintenance, along with stringent control of the pre-analytical, analytical, and post-analytical processes. This comprehensive approach to quality control underscores our commitment to adhering to professional standards and hospital standards, which have been recognized at an international level, providing confidence in the reliability and accuracy of our serum creatinine analysis results.

## Statistical data analysis

Data were analyzed using descriptive and inferential statistics in R software (version 4.3.1) and SPSS computer program version 25.0, following the CLSI EP28 A3c guidelines by the Clinical Laboratory Standards Institute (CLSI) and the International Federation of Clinical Chemistry and Laboratory Medicine (IFCC) [13]. The 95% Reference interval was covered. Age group divisions were assessed using One-way ANOVA; if groups showed no significant differences, they were combined. Each age and gender group were separately assessed, resulting in 20 groups. Data distribution within each group was tested using the Kolmogorov–Smirnov Test and Skewness—Kurtosis test, with results presented in histogram graphs. Outliers were identified using box plots and excluded using Tukey's test, with the Outliers range defined as (Lower Limits (LL) = P25–1.5 x IQR and Upper Limits (UL) = P75 + 1.5 x IQR). Non-Gaussian distributions, after excluding Outliers, were adjusted to Gaussian distribution using the Box-Cox transformation.

## Establishing reference values

Reference values for serum creatinine were established using the Bhattacharya method, a statistical approach with a graphical format. The data needed to be distributed in a Gaussian manner and divided into equal-width frequency bins. Bhattacharya points were determined for each frequency, with the central bin close to the mean or median. A linear consistency (weighted least square) with $R^2 > 0.99$ was sought, using a five-point Savitzky-Golay. The statistical analysis via the Bhattacharya method was carried out using Microsoft Excel Spreadsheet from St Vincent's Hospital (http://www.sydpath.stvincents.com.au/), as depicted in Fig 2.

## Reference value validation

The established reference values were evaluated to show trends in serum creatinine changes from newborn to 18 years using linear graphs. The values for males and females in each age group were compared using % difference = (RIs in males–RIs in females) / RIs in males ×100. The established values were also compared with other studies, including Schlebusch H.'s study in a Caucasian population using the Direct method, Pottel H.'s study in a Caucasian population using the Indirect method (Bhattacharya methods), and Chuang GT.'s study in a

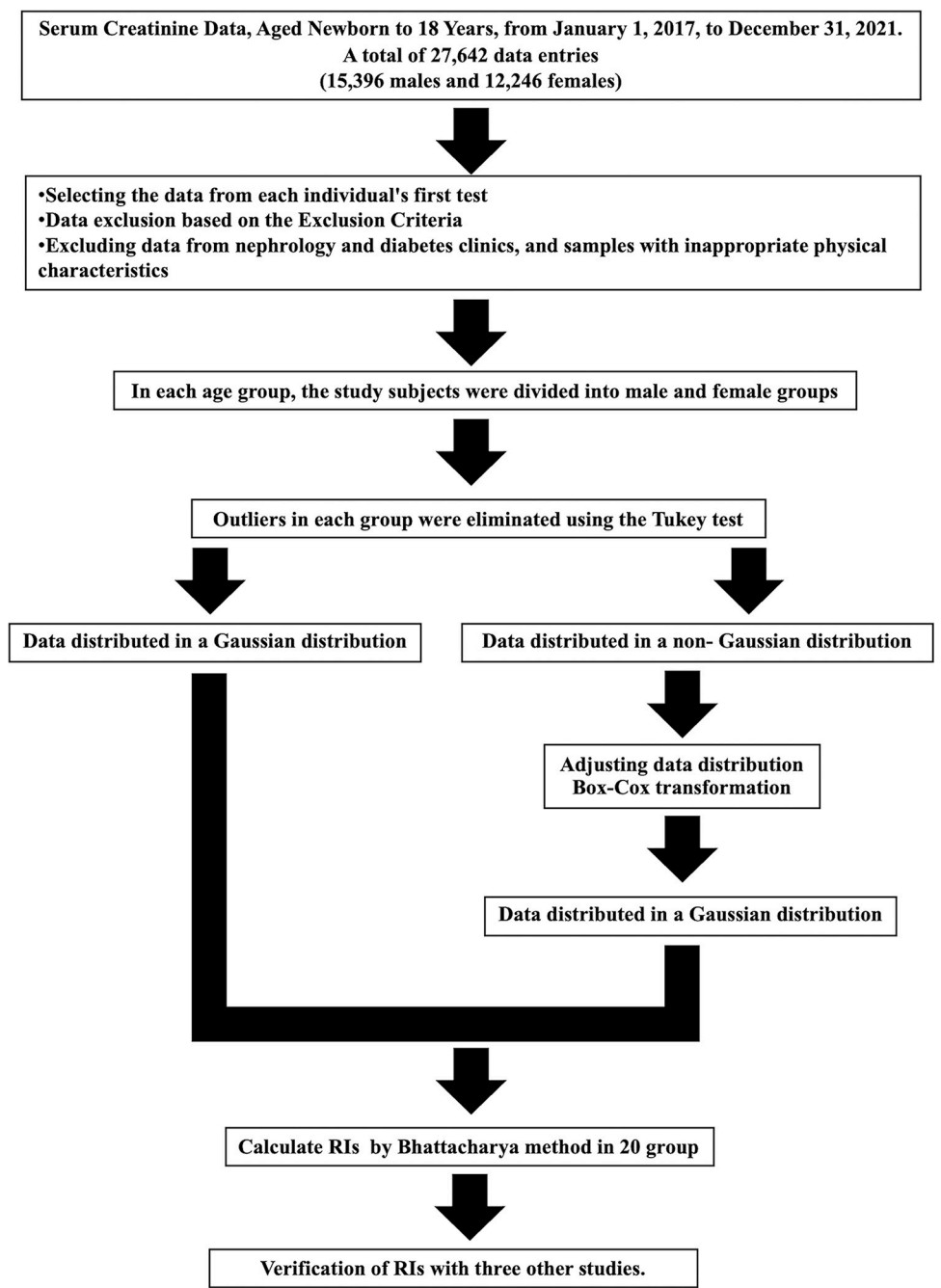

**Fig 1. Steps for establishing reference values from the database of the Queen Sirikit National Institute of Child Health using the Bhattacharya method.**

Taiwanese population using the Indirect method (non-parametric method), presented as % difference (RIs of our studies–RIs of compared studies) / RIs of our studies × 100.

## Ethical considerations in human research

The research was conducted following ethical approvals: Certification from the Director of the Queen Sirikit National Institute of Child Health for using routine laboratory data. Approval from

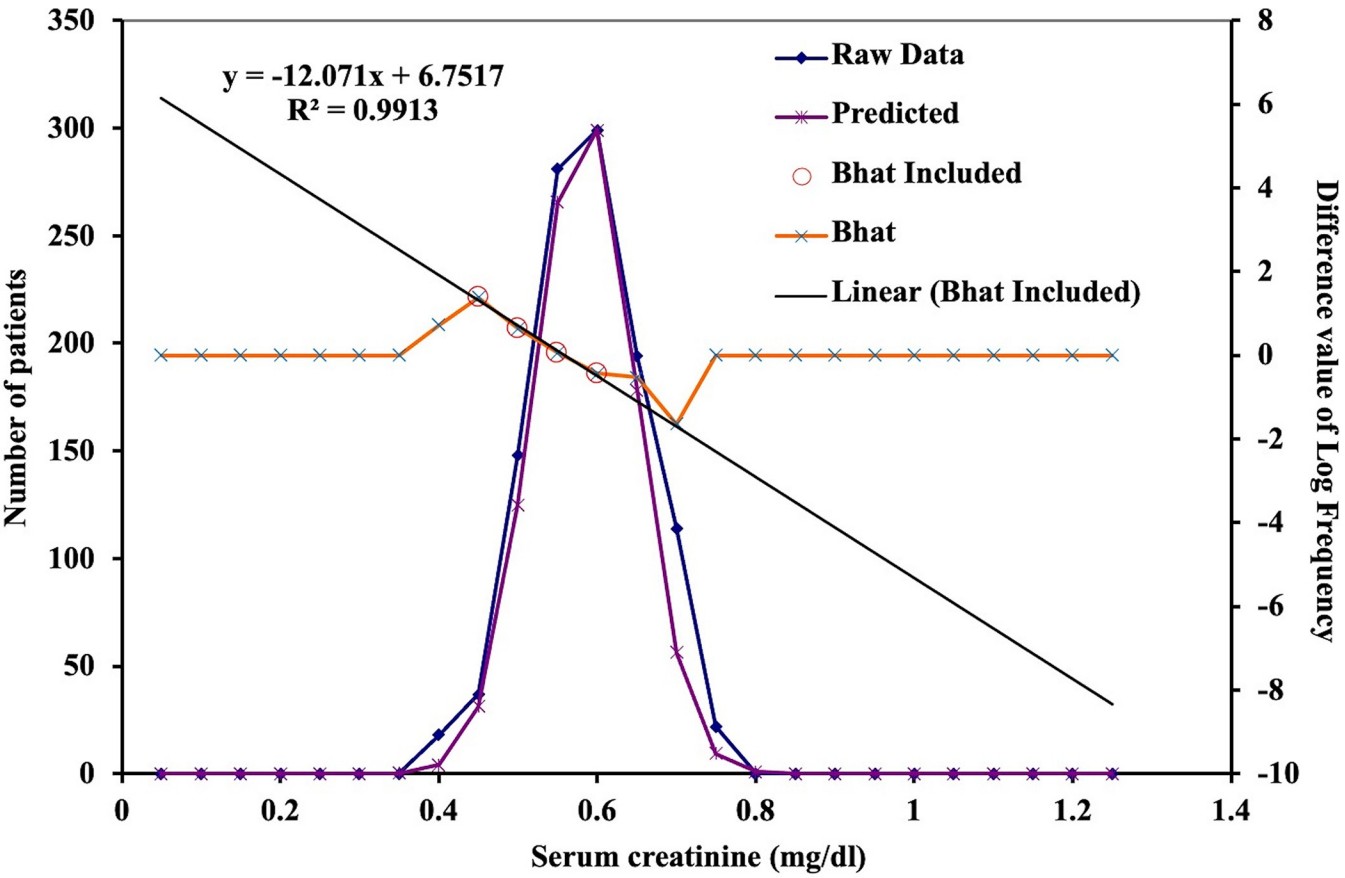

**Fig 2. Statistical methodology for determining reference values using the Bhattacharya method.**

the Queen Sirikit National Institute of Child Health (REC.027/2566) (Exemption) and Human Research Ethics Committee of Thammasat University (Science), (HREC-TUSc) (COE No.018/2566-66AH069). This retrospective study analyzed existing medical records, ensuring patient confidentiality through full data anonymization from January 1, 2017, to December 31, 2021. The Queen Sirikit National Institute of Child Health and Human Research Ethics Committee of Thammasat University (Science), (HREC-TUSc) waived informed consent due to the study's non-interactive nature, aligning with ethical guidelines for retrospective research. The study solely utilized routine laboratory data, with no volunteer involvement or risk of disease condition impact or unintended data exposure. Data included anonymized blood creatinine levels, gender, birth and analysis dates for age calculation, and clinic or ward names. This data was gathered from the LIS system, organized in Microsoft Excel/Microsoft 365 for analysis. The raw data, stored on Thammasat University's OneDrive, was under strict access control. Only Sakon Suwanrungroj, the Principal Investigator, had the authorization to access this data, ensuring stringent data security and confidentiality protocols were maintained. The research commenced only after approval from Human Research Ethics Committee of Thammasat University (Science), (HREC-TUSc), and any modifications to the research protocol required their prior approval.

## Results

The analysis of serum creatinine levels from patients aged newborn to 18 years, spanning from January 1, 2017, to December 31, 2021, yielded 87,480 data points. For reference value

determination, only the first test per individual was used, totaling 29,224 data entries. After applying exclusion criteria, including data from nephrology (1,434 data points) and diabetes clinics (128 data points), and inappropriate sample characteristics (20 data points), 27,642 data entries were selected for reference value establishment, comprising 15,396 males and 12,246 females.

## Statistical data analysis and establishing serum creatinine reference values

The selected data were divided into 10 age groups: (newborn—<2m), (2m-12m), (1- <3y), (3-<5y), (5- <7y), (7- <9y), (9- <11y), (11- <13y), (13- <15y), (15-18y). One-way ANOVA showed significant differences across all 10 groups ($p$-value $\leq$ 0.05). Each age group was further categorized by gender, forming 20 groups in total. After removing outliers, the sample size and descriptive statistics were presented in S1 Table in S1 File. S1 Table in S1 File presents the descriptive statistical data for serum creatinine levels across various age and sex groups in the study. The table includes the number of samples (n), mean, median, standard deviation (SD), and coefficient of variation (CV) for each group. The data show variations in creatinine levels for different age groups, from newborns (0–60 days) to adolescents (15–18 years), and between males and females. Notably, the mean creatinine levels generally increase with age in both sexes, and the variation within each group (as indicated by CV) remains relatively consistent across different age categories. Data were adjusted to fit a Gaussian distribution using the Box-Cox transformation, confirmed by the Kolmogorov–Smirnov Test with a $p$-value $> 0.05$ for ages 2–12 months to 15–18 years. Histograms post Tukey's test and Box-Cox transformation are shown in S1 Fig in S1 File. However, the newborn—<2 months group displayed a non-Gaussian distribution ($p$-value $< 0.05$) in both males and females, even after statistical adjustments.

Table 1, the dataset encompasses a diverse range of age groups, from newborns (0–60 days) to adolescents (15–18 years), with both females (F) and males (M) represented in each group. Each age and sex category includes a substantial number of individuals (n), ensuring robustness in the analysis. The lower reference limit (LRL) and upper reference limit (URL) of serum creatinine levels, measured in mg/dL, were determined for each group. Notably, the reference ranges vary across age groups and between sexes. The results indicate that in the age range of 0–60 days, the reference values are close to those of adults, particularly for females, and the LRL and URL gradually increase in each age group from 2 months to 18 years. For instance, in newborns, the LRL for both females and males is 0.34 mg/dL, while the URLs differ slightly, with females at 1.04 mg/dL and males at 1.06 mg/dL. As age increases, the reference ranges expand accordingly. In the 15–18 years age group, females exhibit an LRL of 0.37 mg/dL and a URL of 0.77 mg/dL, whereas males demonstrate an LRL of 0.50 mg/dL and a URL of 1.00 mg/dL. These findings provide comprehensive reference values for serum creatinine levels in Thai pediatric populations, facilitating accurate diagnosis and monitoring of renal function across various age groups and genders.

Table 2 summarizes the serum creatinine reference values (both lower and upper limits) for Thai children across various age groups from 2 months to 18 years, divided by gender. It shows both the male and female values and calculates the percentage difference between them. The general trend indicates that the differences in lower and upper limits between genders increase with age, particularly noticeable in adolescents (11–18 years). The most significant differences are observed in the 15–18 years age group, with a 26% difference in lower limits and a 23% difference in upper limits between males and females.

## Comparison with other studies

Table 3 shows a comparison of the established reference values with other studies, illustrating the % difference in the upper limits compared to the studies of Schlebusch H., Pottel H., and

**Table 1. Upper and lower limits of serum creatinine reference values (in mg/dL) by Bhattacharya method, categorized by age and gender group.**

| Age | Sex | n | LRL* | URL** |
|---|---|---|---|---|
| 0–60 d*** | F | 1851 | 0.34 | 1.04 |
| | M | 2578 | 0.34 | 1.06 |
| 2–12 m | F | 1334 | 0.15 | 0.34 |
| | M | 1829 | 0.15 | 0.35 |
| 1-<3 y | F | 1811 | 0.18 | 0.40 |
| | M | 2311 | 0.18 | 0.40 |
| 3-<5 y | F | 1280 | 0.22 | 0.46 |
| | M | 1603 | 0.21 | 0.48 |
| 5-<7 y | F | 1117 | 0.24 | 0.52 |
| | M | 1473 | 0.26 | 0.54 |
| 7-<9 y | F | 1113 | 0.29 | 0.57 |
| | M | 1285 | 0.26 | 0.55 |
| 9-<11 y | F | 1076 | 0.29 | 0.60 |
| | M | 1235 | 0.31 | 0.61 |
| 11-<13 y | F | 931 | 0.32 | 0.64 |
| | M | 1155 | 0.34 | 0.71 |
| 13-<15 y | F | 790 | 0.32 | 0.75 |
| | M | 943 | 0.36 | 0.82 |
| 15–18 y | F | 488 | 0.37 | 0.77 |
| | M | 443 | 0.50 | 1.00 |

* LRL = Lower Reference Limits and **URL = Upper Reference Limits 0–60 d*** = non-parametric method

Chuang GT. The studies by Pottel H. and Chuang GT., using the Indirect method, showed negative % differences across all age groups, indicating narrower upper limits than in our study. A line graph comparing serum creatinine reference values between our study and others is shown in Fig 3. In the study by Schlebusch H., reference values were established using the Direct method. It was found that Schlebusch H.'s upper limits were narrower than those determined by the Indirect method in our study, as well as compared to the studies by Pottel H. and Chuang GT. However, it was observed that for the age group of 9 - <15 years in females, the

**Table 2. Reference values for serum creatinine in males and females (in mg/dL) and % difference in reference values between genders across various age groups.**

| Age | LRL* | | | URL** | | |
|---|---|---|---|---|---|---|
| | Male | Female | % Difference | Male | Female | % Difference |
| 2–12 m | 0.15 | 0.15 | 0.00 | 0.35 | 0.34 | 2.90 |
| 1-<3 y | 0.18 | 0.18 | 0.00 | 0.40 | 0.40 | 0.00 |
| 3-<5 y | 0.21 | 0.22 | 4.80 | 0.48 | 0.46 | 4.20 |
| 5-<7 y | 0.26 | 0.24 | 7.70 | 0.54 | 0.52 | 3.70 |
| 7-<9 y | 0.26 | 0.29 | 11.50 | 0.55 | 0.57 | 3.60 |
| 9-<11 y | 0.31 | 0.29 | 6.50 | 0.61 | 0.60 | 1.60 |
| 11-<13 y | 0.34 | 0.32 | 5.90 | 0.71 | 0.64 | 9.90 |
| 13-<15 y | 0.36 | 0.32 | 11.10 | 0.82 | 0.75 | 8.50 |
| 15–18 y | 0.50 | 0.37 | 26.00 | 1.00 | 0.77 | 23.0 |

* LRL = Lower Reference Limits and

**URL = Upper Reference Limits

**Table 3. % Difference in the upper limits of established reference values compared with reference values from the Studies of Schlebusch H et al., Pottel H et al., and Chuang GT et al.**

| Age | Sex | Our study | Schlebusch et al. | | Pottel et al. | | Chuang et al. | |
|---|---|---|---|---|---|---|---|---|
| | | URL | URL | % Difference | URL | % Difference | URL | % Difference |
| 2–12 m | F | 0.34 | 0.39 | -14.70 | 0.36 | -5.90 | 0.39 | -14.70 |
| | M | 0.35 | | -11.40 | | -2.90 | 0.39 | -11.40 |
| 1-<3 y | F | 0.40 | 0.35 | 12.50 | 0.42 | -5.00 | 0.45 | -12.50 |
| | M | 0.40 | | 12.50 | | -5.00 | 0.46 | -15.00 |
| 3-<5 y | F | 0.46 | 0.42 | 8.70 | 0.49 | -6.50 | 0.5 | -8.70 |
| | M | 0.48 | | 12.50 | | -2.10 | 0.51 | -6.30 |
| 5-<7 y | F | 0.52 | 0.47 | 9.60 | 0.54 | -3.80 | 0.56 | -7.70 |
| | M | 0.54 | | 13.00 | | 0.00 | 0.57 | -5.60 |
| 7-<9 y | F | 0.57 | 0.53 | 7.00 | 0.63 | -10.50 | 0.61 | -7.00 |
| | M | 0.55 | | 3.60 | | -14.50 | 0.61 | -10.90 |
| 9-<11 y | F | 0.60 | 0.64 | -6.70 | 0.66 | -10.00 | 0.63 | -5.00 |
| | M | 0.61 | | -4.90 | | -8.20 | 0.68 | -11.50 |
| 11-<13 y | F | 0.64 | 0.68 | -6.30 | 0.76 | -18.80 | 0.73 | -14.10 |
| | M | 0.71 | | 4.20 | | -7.00 | 0.78 | -9.90 |
| 13-<15 y | F | 0.75 | 0.77 | -2.7 | 0.84 | -12.00 | 0.79 | -5.30 |
| | M | 0.82 | | 6.1 | | -2.40 | 0.94 | -14.60 |
| 15–18 y | F | 0.77 | - | - | 0.91 | -18.2 | 0.82 | -6.50 |
| | M | 1.00 | | - | 1.09 | - 9.00 | 1.05 | -5.00 |

URL = Upper Reference Limits; mg/dL

reference values derived from the database of the Queen Sirikit National Institute of Child Health showed narrower upper limits than Schlebusch H.'s study. When comparing the % difference of the upper limits established in our study with those from all three studies, the most significant differences were found in females aged 11 - <13 years and 15–18 years, compared to the study by Pottel H., with our established references being 18.8% and 18.2% narrower, respectively, as shown in the % difference of the upper limits in Table 3.

The comparison of the lower limits of our established reference values with those of Schlebusch H., Pottel H., and Chuang GT. indicated that the lower limits for both males and females in our study tended to be lower than those established for the Caucasian population by Schlebusch H. and Pottel H. but were closer to Chuang GT.'s study in the Taiwanese population. This comparison is depicted in the graph shown in Fig 3.

## Discussion

This study has established reference values for serum creatinine from the Queen Sirikit National Institute of Child Health's database using the Bhattacharya method, categorized by age and gender to provide appropriate references for each age group. However, the age group newborn to 2 months, which was initially segmented, was not included in the reference value determination via the Bhattacharya method. This exclusion was due to the Bhattacharya statistic's requirement for a Gaussian distribution of data [14], which was not achievable for this age group through statistical adjustments. The reference values for serum creatinine in the age range from birth to 2 months are observed to be high and closely align with the reference range for females aged 20 to 40 years [10], and decreased rapidly within the first week, stabilizing in 4 weeks. This pattern is illustrated in S2 Fig in S1 File. The high reference values initially

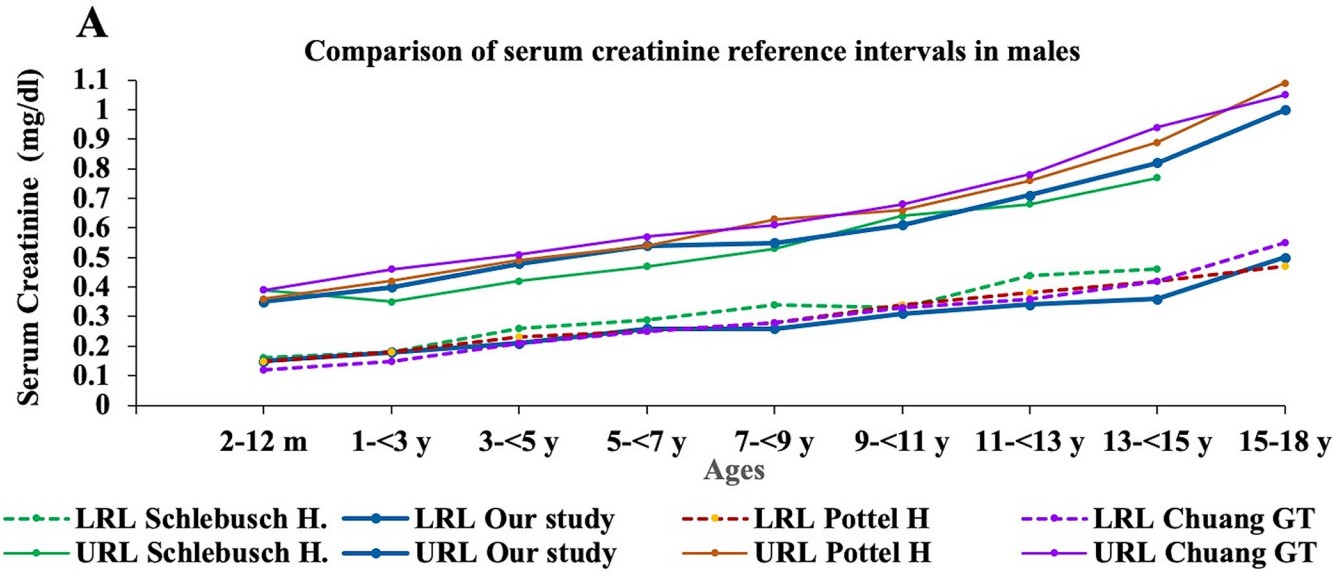

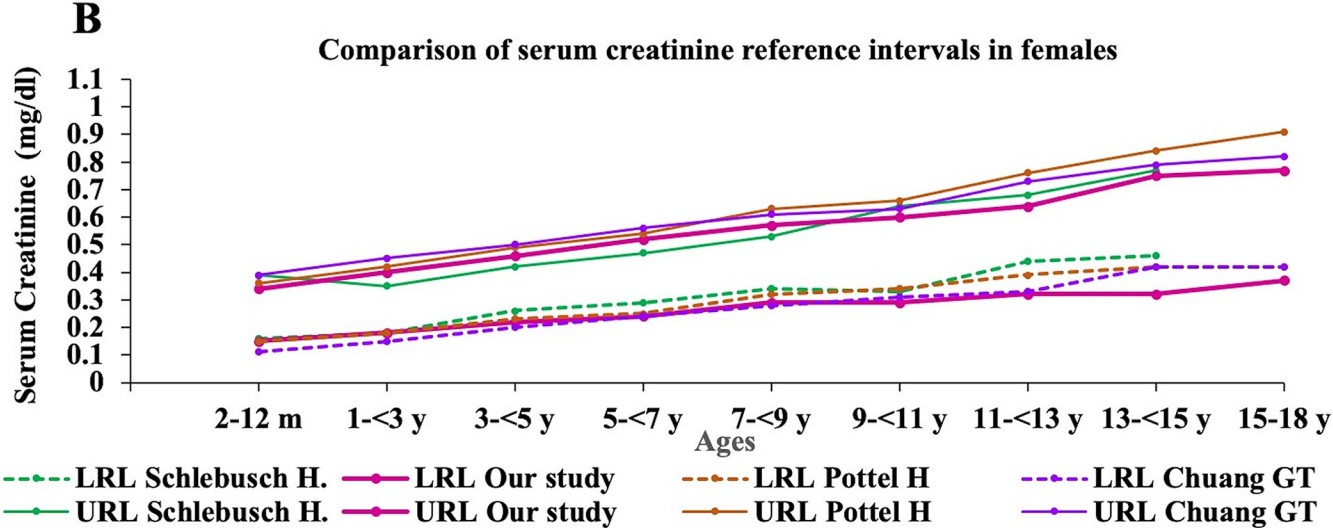

**Fig 3. Graph comparing the upper and lower limits of serum creatinine from established reference values and studies by Schlebusch H., Pottel H., and Chuang GT.** (URL = Upper Reference Limits, LRL = Lower Reference Limits).

are attributed to the newborns having blood components from the mother, resulting in a mixed distribution of two distinct data groups [15], as evidenced by the higher coefficient of variation compared to other groups, detailed in S1 Table in S1 File. Furthermore, due to limitations in data recording and accessibility, it was challenging to separate data for preterm neonates and term neonates. Preterm neonates generally have creatinine values closer to those of females aged 20–40 years, with a broader range of creatinine values compared to term neonates [12]. These factors significantly impacted the data distribution, making it unfeasible to adjust the early newborn data (0–2 months) to fit a normal distribution.

The reference values for serum creatinine established from the Queen Sirikit National Institute of Child Health's database show that from ages 2 months to 18 years, the creatinine

reference values increase with age. This is illustrated in the relationship between serum creatinine values and age from 0 to 18 years in S3 Fig in S1 File. This pattern aligns with the study by Schlebusch H. conducted using the Direct method, a standard approach for establishing reference values currently used in laboratories. The increase in serum creatinine reference values does not differ significantly between males and females in childhood. However, differences become evident during adolescence. Notably, the upper limits begin to diverge around the age of 11- <13 years, with a clear difference in the 15–18 years age group, as depicted in the line graph comparing upper and lower limits of serum creatinine in males and females in S4 Fig in S1 File. This is consistent with previous research findings, including the study by Uemura O. [15] on the Japanese population, which noted significant differences in male and female reference values starting at age 16. The studies by Søeby K. [16] Pottel H. [17] in Caucasian populations found differences beginning at ages 13 and 14, respectively. The rise in creatinine values with age is attributed to the increase in muscle mass [16], and the differences in adolescence are due to the differing muscle mass development in males and females. Males experience a faster increase in muscle mass, influenced by androgen hormones [18], which are produced in greater quantities during adolescence in males and contribute to male physical characteristics and muscle mass development. Consequently, the serum creatinine reference values in males are higher than in females during the onset of adolescence. Additionally, it's noteworthy that meat consumption per capita varies significantly among countries, with Thailand showing relatively lower consumption compared to countries with higher GDPs [19]. This dietary variation is relevant because protein intake from meat can influence serum creatinine levels due to its association with muscle metabolism.

The study involved validating the established serum creatinine values by comparing them with three other studies: Schlebusch H. [12], Pottel H. [17], and Chuang GT. [11]. The upper limit of creatinine is significant for disease diagnosis and treatment assessment, as it indicates abnormalities in kidney function. Continuously rising creatinine levels up to the upper reference limit may signal kidney dysfunction [20]. In the comparison with the three studies, it was found that our established reference values have a lower upper limit than those of Pottel H. and Chuang GT. across all age groups. Both these studies determined reference values using the Indirect method, similar to our approach. Pottel H.'s study, which set reference values for a Caucasian population using the Bhattacharya method, showed wider upper limits compared to ours. This discrepancy can be attributed to population differences, as ethnicity is a factor influencing creatinine values [2]. Additionally, variations in muscle mass across populations may contribute to these differences. Chuang GT.'s study, conducted with a Taiwanese (Asian) population, presented higher reference values, which were consistently above ours, as shown in Fig 3. This difference might be due to the methodology used in establishing reference values.

Chuang GT.'s study established reference values using a non-parametric method, setting the lower and upper limits at the 2.5th and 97.5th percentiles, respectively. However, our reference values were determined using the Bhattacharya method, recommended for establishing reference values through the Indirect method [6]. Yang C.'s study [21] compared reference values established from the same database using different methods and found that those determined by the Bhattacharya method had more effective upper limits than the non-parametric method. In the study by Chuagn GT., Jaffe's method was employed for creatinine analysis, noted for its cost-effectiveness yet prone to interferences, potentially affecting the specificity of the results. This susceptibility may contribute to the broader creatinine values observed in Chuagn GT.'s study, compared to ours. Contrasting this, Schmidt RL's study [22] evaluated creatinine levels using both Jaffe's and the Enzymatic methods, finding negligible differences in outcomes, suggesting that the methodological choice might not significantly impact the

comparability of creatinine values across studies. For the reference values in Schlebusch H.'s study, the Direct method was utilized, considered a standard approach for establishing reference values as recommended by the CLSI EP28 A3c [13]. Schlebusch H. conducted the study in the age range of 0–15 years. Therefore, the comparison of the reference values established in our study with those of Schlebusch H. was only up to the age range of 13 - <15 years. The comparison revealed that the upper limits from Schlebusch H.'s study were lower and had a narrower range than those of other studies established using the Indirect method, including our study. However, it was observed that in the age range of 9–15 years in females, the upper limits from our study were found to be narrower than those in Schlebusch H.'s study, as depicted in Fig 3. This difference might be attributed to varying population characteristics, leading to distinct muscle mass development patterns in adolescence.

Upon examining Schlebusch H.'s study, it was noted that the sample size for the age group starting from 11 years was smaller compared to other age groups. This could be a factor contributing to the broader range of values in this age segment. However, considering the differences in muscle mass, literature reviews have yet to clearly demonstrate a comparative analysis of muscle mass characteristics or quantities among pediatric populations of Caucasian and Asian descent. Nevertheless, Jensen B.'s study [23] in a population aged 18–78 years found that Caucasians have a greater muscle mass than Asians, corroborating with ABE T.'s study [24], which showed that the skeletal muscle mass (SMM) of Asians was lower than that of Caucasians and Brazilians. For the comparison of lower limits of serum creatinine in the 2–12 months age range, the comparison with the studies of Pottel H. and Chuang GT. was not displayed due to the different age ranges studied (1–12 months), leading to a discrepancy in age categorization. However, upon comparing the lower limits with these three studies, it was found that the reference values we established tend to be lower than other studies but are more closely aligned with Chuang GT.'s study in the Asian population. This variation in lower limits could be attributed to differences in population characteristics and muscle mass quantities, leading to lower limits in our study compared to those established in Caucasian populations. This is further elaborated in S2 Table in S1 File, which outlines the percentage differences in the lower reference limits when our findings are compared to the studies conducted by Schlebusch H., Pottel H., and Chuang GT.

This study aimed to establish reference values, stratified by age and gender, using data from the Queen Sirikit National Institute of Child Health. It seeks to provide age- and gender-appropriate creatinine reference values for Thai children. The findings indicate that current reference values for creatinine can effectively monitor renal abnormalities or the onset of renal degeneration. However, particular vigilance is warranted in females aged 9–15 years, as the study revealed narrower upper limit reference values compared to those currently in use. Implementing these newly established reference values for Thai children could lead to earlier detection and monitoring of renal abnormalities, ultimately enhancing the accuracy of disease diagnosis and prognosis.

Although the creatinine reference values derived from the Queen Sirikit National Institute of Child Health's database are suitable and specific for Thai children, aiding physicians in precise kidney function assessments, serum creatinine levels are still influenced by various factors such as gender, ethnicity, age, muscle mass, diet, and physical activity [2]. Therefore, changes in creatinine levels are not solely indicative of kidney function efficiency and can affect the assessment and interpretation of creatinine values. Currently, considering Cystatin C as an alternative measure for kidney function evaluation is advised. Cystatin C, a single-stranded polypeptide, can be freely filtered through the glomerulus and is fully reabsorbed and catabolized by proximal tubular cells. Importantly, Cystatin C is not influenced by age, gender,

weight, height, or other bodily changes, making it a more specific indicator for kidney function efficiency and allowing for easier and more accurate identification of abnormalities [25].

In future studies, expanding the dataset to include data from other analyzers or test sites within Thailand could enhance the generalizability of our reference intervals. Incorporating data from diverse sources helps mitigate potential errors and biases inherent to single sites or instruments, thereby strengthening the reliability and applicability of our findings across various laboratory platforms. Additionally, exploring the feasibility of collaborative efforts with multiple healthcare institutions to collect and analyze serum creatinine data could further enrich our understanding of age and gender-specific variations in the pediatric population.

## Conclusion

This study established reference values using the database from the Queen Sirikit National Institute of Child Health, applying the Bhattacharya method and categorizing by age and gender, to create appropriate references for Thai children. The results show changing reference ranges across all age groups, with both upper and lower limits gradually increasing from 2 months to 18 years. The analysis of gender-specific reference values reveals similarities in childhood but distinct differences as children enter adolescence. These newly established reference values are a crucial tool for laboratories, aiding physicians in assessing kidney abnormalities specific to Thai children of each gender and age group. This leads to quicker and more accurate diagnoses and treatments, greatly benefiting the Thai pediatric population.

## Research limitations

This study, utilizing an existing database, faced limitations in data access, which affected the separation of premature and full-term birth histories. Consequently, we couldn't distinguish these two groups in the newborn to <2 months age range. Additionally, the completeness and accuracy of the recorded data might not be entirely reliable. This means that the study, employing the Indirect method, might include patient data in the analyzed dataset. Even though data selection was conducted with comprehensive inclusion and exclusion criteria, and aimed to represent a healthy population, these limitations could impact the study's findings.

## Supporting information

**S1 File.**
(PDF)

## Acknowledgments

The authors gratefully acknowledge the research and innovation support provided by the Thammasat University Research Unit in Medical Technology and Precision Medicine Innovation.

## Author Contributions

**Conceptualization:** Sakon Suwanrungroj.

**Data curation:** Sakon Suwanrungroj.

**Formal analysis:** Sakon Suwanrungroj, Parichart Pattarapanitchai.

**Investigation:** Sakon Suwanrungroj.

**Methodology:** Sakon Suwanrungroj, Parichart Pattarapanitchai.

**Project administration:** Sirinart Chomean.

**Resources:** Sirinart Chomean.

**Supervision:** Parichart Pattarapanitchai, Chollanot Kaset.

**Validation:** Parichart Pattarapanitchai, Chollanot Kaset.

**Visualization:** Chollanot Kaset.

**Writing – original draft:** Sakon Suwanrungroj.

**Writing – review & editing:** Sirinart Chomean, Chollanot Kaset.

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
