## [Decision Letter · Decision Letter 0]

14 Feb 2024

PONE-D-23-42297Establishing Age and Gender-Specific Serum Creatinine Reference Ranges for Thai Pediatric PopulationPLOS ONE

Dear Dr. Kaset,

Thank you for submitting your manuscript to PLOS ONE. After careful consideration, we feel that it has merit but does not fully meet PLOS ONE’s publication criteria as it currently stands. Therefore, we invite you to submit a revised version of the manuscript that addresses the points raised during the review process.

Reviewer 1

Page 4, lines 82-87

Can the exclusion criteria be more stringent? What was the general health status of the subjects? If they were healthy, how do you define or verify that?

Page 4, lines 65-57

How about conditions that could have indirectly impacted on renal function e.g. severe dehydration?

Page 5, lines 99-104

Good to see QC activities to reduce analytical errors. What considerations were made for the minimization of the inclusion of results associated with pre-analytical errors?

Page 10, lines 197-198

Results on age group (newborn - < 2months were not presented

Page 15, lines 272-275

You could make a more confident claim if you used inferential statistics to evaluate this observation

Page 16, lines 296-299

Could it also be due to dietary differences? e.g. which of these countries have a higher meat consumption per capita?

Reviewer 2

Establishing Age and Gender-Specific Serum Creatinine Reference Ranges for Thai Pediatric Population

Introduction

Creatinine, a widely used renal biomarker, aids in diagnosing and monitoring acute and chronic kidney diseases. However, factors like muscle mass, sex, ethnicity, and age influence its levels. This study addresses this crucial gap by establishing reference intervals for Thai children and adolescents (under 18 years).

The subject is clearly explained with a strong background highlighting the study's significance. The research aim is also explicitly stated.

Methodology

Retrospective data from the Queen Sirikit National Institute of Child Health (January 1, 2017, to December 31, 2021) was used. The analysis included 29,224 entries, representing the first creatinine test per individual, avoiding data duplication issues.

The sample size, with sufficient data points, is appropriate for reference interval studies. Serum creatinine analysis employed the well-established enzymatic colorimetric method on the COBAS INTEGRA 400/400 PLUS analyzer. Both internal and external quality control data were presented, ensuring the reliability of reference value interpretation.

Including data from other analyzers or test sites within Thailand could strengthen the generalizability of the reference intervals, as they will be applied across various platforms. Data from diverse sources helps eliminate errors and biases inherent to single sites or instruments.

Results/Analysis

Results are clearly presented in easy-to-understand tables and charts, categorized by age group and gender for improved analysis. Graphs and figures are well-labeled and comprehensible.

Outliers were rightfully excluded, and non-Gaussian data was normalized using standard statistical tools.

However, the very large SD/CV for male 5-7 year olds' serum creatinine results requires explanation. It might bias the reference range for that age group.

General Remarks

This well-written paper clearly presents the background, aim, methodology, and results. The researchers acknowledge the study's limitations, and further improvement in accuracy and scope using different statistical tools is encouraged.

We look forward to receiving your revised manuscript.

Kind regards,

Emmanuel Kwaku Ofori, PhD

Academic Editor

PLOS ONE

“Thammasat University Research Unit in Medical Technology and Precision Medicine Innovation.”

Reviewers' comments:

Reviewer's Responses to Questions

**Comments to the Author**

1. Is the manuscript technically sound, and do the data support the conclusions?

Reviewer #1: Yes

Reviewer #2: Yes

2. Has the statistical analysis been performed appropriately and rigorously? 

Reviewer #1: Yes

Reviewer #2: Yes

3. Have the authors made all data underlying the findings in their manuscript fully available?

Reviewer #1: No

Reviewer #2: Yes

4. Is the manuscript presented in an intelligible fashion and written in standard English?

Reviewer #1: Yes

Reviewer #2: Yes

5. Review Comments to the Author

Reviewer #1: The authors performed One-Way ANOVA test for the establishment of the groups for the establishment of the reference ranges. However, the result for that analysis was not presented.

This manuscript requires some minor revisions.

Reviewer #2: Establishing Age and Gender-Specific Serum Creatinine Reference Ranges for Thai Pediatric Population

Introduction

Creatinine, a widely used renal biomarker, aids in diagnosing and monitoring acute and chronic kidney diseases. However, factors like muscle mass, sex, ethnicity, and age influence its levels. This study addresses this crucial gap by establishing reference intervals for Thai children and adolescents (under 18 years).

The subject is clearly explained with a strong background highlighting the study's significance. The research aim is also explicitly stated.

Methodology

Retrospective data from the Queen Sirikit National Institute of Child Health (January 1, 2017, to December 31, 2021) was used. The analysis included 29,224 entries, representing the first creatinine test per individual, avoiding data duplication issues.

The sample size, with sufficient data points, is appropriate for reference interval studies. Serum creatinine analysis employed the well-established enzymatic colorimetric method on the COBAS INTEGRA 400/400 PLUS analyzer. Both internal and external quality control data were presented, ensuring the reliability of reference value interpretation.

Including data from other analyzers or test sites within Thailand could strengthen the generalizability of the reference intervals, as they will be applied across various platforms. Data from diverse sources helps eliminate errors and biases inherent to single sites or instruments.

Results/Analysis

Results are clearly presented in easy-to-understand tables and charts, categorized by age group and gender for improved analysis. Graphs and figures are well-labeled and comprehensible.

Outliers were rightfully excluded, and non-Gaussian data was normalized using standard statistical tools.

However, the very large SD/CV for male 5-7 year olds' serum creatinine results requires explanation. It could be a typographical error, but if not, it might bias the reference range for that age group.

The WHO-recommended Bhattacharya method established reference values, and the upper and lower limits were compared to three similar studies in this age population.

General Remarks

This well-written paper clearly presents the background, aim, methodology, and results. The researchers acknowledge the study's limitations, and further improvement in accuracy and scope using different statistical tools is encouraged.

6. PLOS authors have the option to publish the peer review history of their article (what does this mean?). If published, this will include your full peer review and any attached files.

Reviewer #1: No

Reviewer #2: **Yes: **Kumahor Elikem

---

## [Author Response · Author response to Decision Letter 0]

23 Feb 2024

Thank you for dedicating your valuable time to reviewing our manuscript. Your thoughtful insights and constructive feedback are greatly appreciated.

---

## [Editor Report · Decision Letter 1]

27 Feb 2024

Establishing Age and Gender-Specific Serum Creatinine Reference Ranges for Thai Pediatric Population

PONE-D-23-42297R1

Dear Dr. Kaset,

We’re pleased to inform you that your manuscript has been judged scientifically suitable for publication and will be formally accepted for publication once it meets all outstanding technical requirements.

Kind regards,

Emmanuel Kwaku Ofori, PhD

Academic Editor

PLOS ONE
---

## [Editor Report · Acceptance letter]

29 Feb 2024

PONE-D-23-42297R1 

PLOS ONE

Dear Dr. Kaset, 

I'm pleased to inform you that your manuscript has been deemed suitable for publication in PLOS ONE. Congratulations! Your manuscript is now being handed over to our production team.

Kind regards, 

on behalf of

Dr. Emmanuel Kwaku Ofori 

Academic Editor

PLOS ONE